# The Influence of Social Capital on Protective Action Perceptions Towards Hazardous Chemicals

**DOI:** 10.3390/ijerph17041453

**Published:** 2020-02-24

**Authors:** Tiezhong Liu, Huyuan Zhang, Hubo Zhang

**Affiliations:** 1School of Management and Economics, Beijing Institute of Technology, Beijing 100081, China; liutiezhong@bit.edu.cn (T.L.); 3120160694@bit.edu.cn (H.Z.); 2China Electronics Standardization Institute, Beijing 100007, China

**Keywords:** protective action perceptions, risk perception, social capital, hazardous chemicals

## Abstract

The stigmatized character of hazardous chemicals has caused individuals in hazards to take excessive protective actions. Here, social capital is introduced to discuss its influence on the protective action decision model (PADM), considering this variable has a relatively high individual trust level in regards to information on hazardous chemicals. A model was constructed by taking protective action perceptions as the dependent variable, social capital as the independent variable, the pre-decision process as the mediating variable, and socioeconomic status as the moderating variables. Data were collected with a neighborhood sampling method, and a total of 457 questionnaires were obtained from neighboring residents near a large cold ammonia storage house in Haidian District, Beijing. Results: While the family and friendship networks produced a larger positive influence, the kinship network produced a smaller positive influence; furthermore, the influence of social capital must be brought through the pre-decision process; finally, socioeconomic status has a directional moderation on the friendship network, an enhancing moderation on the kinship network, and a weakening moderation on the family network.

## 1. Introduction

Rapid urban expansion, coupled with the lack of an integrated urban construction plan, has led residential communities to be gradually surrounded by chemical enterprises sprawling about China. Hence the term “the city surrounded by the chemical industry” [1] has been coined particularly to address the severity. However, once hazardous chemicals installation fails, it brings a serious threat to the ecological environment and human health [2,3,4]. The stigmatized character of hazardous chemicals [5] has created immense communicational problems on hazard information between authorities and residents. Facing these problems, either insufficient protective actions or excessive protective actions have been taken at the same time by nearby residents who may have got either inaccurate or distorted information. For example, the explosion accident of Qingdao Oil Pipeline on 22th November 2012, caused the death of many passers-by and employees in the surrounding units or communities. Also, the number of protests against PX chemical projects has been constantly surging across mainland China since [6]. Therefore, it has become a daunting task for cities with dangerous chemical enterprises to carry out research on risk communications of chemical products.

Reliable information sources are the key foundation for protective action perceptions [7], the social network formed by family, friends, and kinship serve as important channels from which individuals obtain risk information [8,9,10,11]. Because the hazardous chemicals are technically complex hazard sources, the public is often placed at an unfavorable position, which obstructs them from getting relevant information on hazardous chemicals in time. A lot of information on hazardous chemicals are collected by the public through their trusted information channels [12,13,14,15]. Furthermore, the type of social network can affect an individual’s actions. For example, families who have children may decide to relocate because of the existence of nearby chemical enterprises. However, the specific content involving how social relations affect risk communications still need to be discussed in depth [8,16]. Until now, the concept of social capital has been closely connected with social relations, which is a kind of capital that can elevate individuals’ social reputation, strengthen mutual trust among individuals, increase values for the informal norms of the whole groups or organizations and bring returns to individuals [17,18]. Also, social capital can affect the process of information transmission, communication of human emotions, trust built-up, and generation of social certificates, thus exerting great influence on all walks of life of this “relational society” [19,20]. As the construction of trust relationships plays a critical role in the risk communication of hazardous chemicals, it is necessary to study the influence of social capital on risk communication of hazardous chemicals.

In summary, the study of social capital on public risk perception can help obliterate problems of risk communication of hazardous chemicals. A search scheme was constructed as described below: In view of the research background of hazard risk communication of hazardous chemicals, the residents around the installation of hazardous chemicals have been selected as the sample frame. In particular, the protective action decision model (PADM) was introduced in line with the research idea that social capital affects protective action perceptions by first affecting psychological behaviors. In the paper, the social relations are quantified with the theory of social capital; then, a model has been constructed which can help explain the influence of social capital on the risk perceptions of hazardous chemicals protective actions. Later, the result will be used in tackling the practical problems of risk communication of hazardous chemicals.

## 2. Materials and Methods 

### 2.1. Theoretical Framework

The protective action perceptions of surrounding residents of hazardous chemicals were taken as the dependent variable, as well as the social capital as the independent variable. With consideration of the mediating effect of the pre-decision process and the moderating effect of socioeconomic status, the influence model of social capital on protective action perceptions has been constructed, as shown in Figure 1.

The protective action perceptions, which refers to the perception level of individuals on taking response actions against potential hazards, was selected as the dependent variable [21]. The concept presents a comprehensive reflection of individual psychological factors and ability factors [22]. To be more specific, the psychological factors refer to the level of trust in sources of advice on relevant protective actions and professional evaluation; the ability factors refer to individual risk response knowledge, skills, and resources etc. It was discovered by relevant studies that [23,24,25]: the protective action perceptions can be affected by risk information in two manners, the hazardous consequence to life and health, and property losses. That is, if a hazardous chemical accident occurs, a factor related to the possibility of evacuation, emergency protective measures taken by surrounding residents would be used in the measurement of the protective action perceptions. For example, the terms “evacuation behavior” and “emergency protective measures” have been taken particularly to measure this variable. While evacuation behavior is related to emergency response resources (such as vehicles), the emergency protective measures involve the usual protective resources (such as fortification of houses). In other words, the former focuses more on the resources of short-term disaster responses; the latter focuses more on the resources of long-term hazard adjustments.

As the most important factor affecting the protective action perceptions, the concept of social capital was chosen as the independent variable for this study. In particular, the concept of social capital is divided into three types, based on the particular characteristics of relation networks in China [26,27]: family network, kinship network, and friendship network. First of all, family connections relate to social relations among family members, such as the relationship between husbands and wives, parents and children.To elaborate, the indicators of family networks generally involve “family responsibility, family status and family scales” and other indicators [28].In definition, “family responsibility” refers to a sense of responsibility to family members, which could influence the behavior; “family status” refers to the individual’s speech right in decision-making in their own families; the “family scale” refers to the number of people living together.Secondly, the kinship connections refer to social relations formed by relatives. In terms of the kinship network, some characteristics of the kinship network can generally be described as “kinship distances, intimacy, the scale of kinship”, and other indicators [29]. To be more exact, “kinship distance” refers to the geographic distances between relatives; “intimacy” refers to the extent of alienation between relatives, “scale of kinship” refers to the number of frequent relatives. Finally, the connections of friends point to friendships or cooperation among individuals. Specifically, the friendship network is generally characterized by such indicators as “closeness, trust, heterogeneity, upper reachability, and friendship scale” [30]. In particular, “closeness” refers to the frequency of interaction between friends; “trust” refers to the level of mutual trust between friends; “heterogeneity” refers to the occupational and educational differences between friends; “upper reachability” refers to the ability to obtain the best resources from friends; “friend scales” refers to the number of frequent friends.

Next, for more details on the pre-decision process, no matter how risk perception is affected by hazard risk information, individuals still play an important part in affecting the process of information reception, clue detection and understanding of the risk clues [31,32]. Hence, the term pre-decision process variable was created by Lindell in his PADM model [8]. Meanwhile, the pre-decision process was chosen as a mediating variable.

At last, as demonstrated by the subject of demography, socioeconomic status is perceived as an important influencing factor for individual protective actions. For example, it is found that families with higher income levels have a bigger proportion of insurance purchases [33]. With this in mind, “socioeconomic status” was chosen as the moderating variable.

### 2.2. Hypothesis

#### 2.2.1. The Hypothesis on Direct Influence

The process of decision-making of protective actions is determined by the emotional factors of individuals, the individuals’ abilities to understand hazard information, and the applicable resources, etc. Social capital, as the resource for the family network, kinship network, friendship network and other social networks around individuals, can be perceived as having a significant effect on the abilities and emotions of individuals on information acquisition and protective action perception. As a result, hypothesis 1 was raised as:

Social capital directly affects the protective actions perceptions of hazardous chemicals.

#### 2.2.2. The Mediating Variables and Hypothesis on Mediating Actions

The behavioral influence can be produced only after the hazard information is processed by the factor of individual psychology [34,35]. In other words, the influence of protective action perceptions would be produced only when information, resources, and emotions brought to individuals through social capital are received, attended to, and understood clearly [8]. The terms such as “acceptance”, “attention” and “understanding” were selected specifically as the measurement indicators for the pre-decision process. To elaborate, “acceptance” refers to relevant information acquisition, “attention” refers to the extent of attention attached to the information, “understanding” refers to the processing abilities on current information and opinion generation abilities. Thus, hypothesis 2 was proposed accordingly as:

The pre-decision process plays a mediating effect on the perception of social capital and hazardous chemicals.

#### 2.2.3. Moderating Variables and Hypothesis on Moderating Effect

Socioeconomic status reflects the social evaluation of the comprehensive values of individuals. It is found for example, that the physical health risks and social risks are different among people with different socioeconomic statuses; the distribution of environmental risk is reversely overlapping with wealth and power distribution to some extent [36]. Also, education, income, and other socioeconomic status factors could have an important impact on the relationship between social capital and pre-decision process: As for individuals of similar social capital structures, the higher their education level is, the more capable they are in extracting social capital resources and taking practical actions; the higher their income level is, the more they would focus on the safety of their own property [37,38]. It can be said therefore that, different socioeconomic status can have different influences on the protective action perceptions of urban residents. Thus, hypothesis 3 was raised as:

The socioeconomic status can moderate the relationship between social capital, and the pre-decision process put forward.

### 2.3. Survey

#### 2.3.1. Questionnaire

A questionnaire was used to collect data based on research variables and hypothesis which was designed with two steps. As a first step, a team was assembled by teachers and students, and then the analysis would be conducted based on the conceptual model. Later, the preliminary draft would be laid out after two months of professional discussion. In the second step, the content of the questionnaire was finalized after three rounds of amendments by experts in fields of hazardous chemicals and risk communication. The final questionnaire, in which the Likert 5-Scale was adopted, consisted of four aspects as the “demographic information (including socioeconomic status)”, “social capital”, the “pre-decision process”, and the “protection action perceptions”.

#### 2.3.2. Sampling

For sampling, residents of the “Great Bell Temple” (GBT) community were chosen along with one typical installation of hazardous chemicals”. The community is adjacent to a large cold storage house, where a large amount of liquid ammonia was used to freeze the fresh meat. Before the interview, investigators had gone through a series of intensive pieces of training, which included such training aspects as necessary interview techniques, sample psychological knowledge, and integrity. The scale of the sample was calculated in terms of a finite population [39]: $N\geq \frac{t_{a}^{2}*p*q*N}{\left(N-1\right)*e^{2}+t_{a}^{2}*p*q}$. Before the scale of the sample was calculated, several parameters were determined: n = sample scale to be calculated; N = scale of the population living in GBT community from which the sample is drawn; p = expected percentage of the response variable, which is 80%; also, q = 1 − p; e = accepted margin of error, which is 5%; $t_{a}$ = 1.96. In addition, the sample scale can be obtained with the above principle: n ≥ 237.

The survey was implemented using the method of neighborhood sampling [40], that is: the survey was carried out from 16:00 to 21:00 on three consecutive weekends to ensure the survey had taken answers from all family members. To begin with, one household was selected inside the community randomly, after that the nearest door would be selected, the survey was not stopped until we obtained enough residents; then, the residents were interviewed with questionnaires prepared in advance; however, the survey would be abandoned if no one is available for the survey after contacts are made three times. Through the above survey steps, it could not only ensure the maximum coverage of all families in the community but also improve the survey efficiency.

#### 2.3.3. Data

Then 650 questionnaires were collected. In which 457 were identified as valid questionnaires, with an effective recovery of 70.3%. The distribution of respondents are as follows: In terms of gender, 49% were male and 51% were female; in terms of age, 44% were aged between 0–29, 46% were aged between 30–45, and 10% were aged 45 and above; in terms of education, 12% participants received a high school education or below, 70% were undergraduates, and 18% were graduates.

The reliability and validity of the processed data were analyzed. The coefficient of Cronbach’s α was measured as 0.814, which indicates that the reliability of the questionnaire had met the requirements [41]. KMO (Kaiser-Meyer-Olkin) was measured as 0.826, and the significant *p*-value was measured as 0.00, which testified that the requirement for the validity had been met. In terms of the processing of abnormal data: directly eliminating those that could be spotted by experience and common sense while replacing those that were found through the stem and leaf display as a result of the mean difference method. In terms of the processing of the missing value: directly delete questionnaires with 20% or more missing information, and process the missing information of the remaining questionnaire with the hot deck filling method [12,42].

## 3. Results

### 3.1. Statistical Analysis

First of all, three kinds of social capital independent variables, family network, kinship network, and friend network, were defined and statistically analyzed. Then, the correlation between them and two kinds of measurement variables, namely ”the evacuation behavior and emergency protective measures”, were analyzed to reveal the basic characteristics of the research variables.

#### 3.1.1. The Family Network Variables

The descriptive statistics reflected that: the average scale of the family network was 3.47, that is, there were 3.47 people in each respondent’s family, with a standard deviation of 1.037, indicating that the respondent’s family is basically a small scale family; also, the average value of family responsibility was measured as 3.89, indicating that the responsibility was high for the family of the respondent; the average value of the family status of the respondent was 3.99, indicating that the respondent possesses high decision-making power or discourse power in the family. The correlation analysis showed that: To begin with, the family responsibility was positively correlated with evacuation behavior; second, there was a positive correlation between family scale, family responsibility, family location, and emergency protective measures.

#### 3.1.2. The Kinship Network Variables

It could be seen from descriptive statistics that: The average scale of kinship network was 3.17, that is, each respondent had about 5.84 frequently contacted relatives, with a standard deviation of 0.728, indicating that there were few connections between respondents and relatives; the average kinship distance was measured as 3.31, with a standard deviation of 1.199, indicating that the respondents and relatives may be kept at a relatively long distance, but with much difference; the figure for average intimacy was 3.21, reflecting that the respondents and their relatives were loosely connected. The correlation analysis showed that: Intimacy was positively correlated with evacuation behavior and was positively correlated with emergency protective measures.

#### 3.1.3. The Friendship Network Variables

The following could be said about the descriptive statistics: The average scale of the friendship network was 3.52, i.e., every respondent had about 6.54 regular friends, with a standard deviation of 0.937, indicating that there was not much contact between the respondents and their friends. The closeness average was 3.85, i.e., there was a strong connection between the respondents and their friends, or a strong relationship. The average value of trust was 4.07, i.e., the respondents trusted their friends very much. The mean value of heterogeneity was 2.98, which indicates that few differences were found in occupation, hobbies, or other aspects of the frequently contacted friends of the respondents. The mean value for the upper reachability was 3.39, i.e., there was little difference in resources, information and other aspects between the respondents and their frequently contacted friends. The correlation analysis showed that: to begin with, “trust, heterogeneity, and upper reachability” were positively correlated with evacuation behavior; second, “closeness, trust, heterogeneity, and upper reachability” were positively correlated with emergency protective measures.

### 3.2. Modeling Analysis

Based on the conceptual model, a modeling analysis was conducted on the structural equation model (SEM) which involves: the dependent variables of protective action perceptions, independent variables of social capital, mediating variables of the pre-decision process, and moderating variables of socioeconomic status. The model was constructed with the following considerations: At first, based on whether the variables should be adjusted, three or four factors models were constructed. As to the three factors model, the influence of mediating variables was considered. The analysis of the influence paths of “social capital—pre-decision process—protective action perceptions” is displayed in Figure 2a. As to the four factor model, both the role of the influence of mediating variables and moderating variables were considered. The analysis of the influence path of “social capital—pre-decision process—socioeconomic status—protective action perceptions” is shown in Figure 2b. Eventually, six types of influence were obtained through the combination between “family network”, “kinship network”, “friendship network”, and three or four factor models.

From the data analysis, three factors were not significant (t ≤ 1.96), which were “family scale” in the family network, “kinship scale” in the kinship network, and “friend scale” in the friend network, so they were not considered in the analysis. In other words, only factors of “family responsibility” and “family status” were considered as the measurement indicators for the family network; only two indicators of “relative distance and intimacy” were considered as the measurement indicators for the kinship network; only four indicators of “closeness, trust, heterogeneity and upper reachability” were considered as the measurement indicators for the friendship network. Based on the analysis using amos20.0 software (IBM, Armonk, USA), the output value of goodness of fit index was obtained, as shown in Table 1.

It can be seen from the Table 1 on the goodness of fit index of SEM, all CMIN/DFs (Chi Square Degree of Freedom) were less than 3, all CFIs (Comparative Fit Index) were greater than 0.9, all RMSEAs (Root Mean Square Error of Approximation) were less than 0.060, all other goodness of fit index also met the requirements, which testify to the applicability of the modal for the analysis.

### 3.3. Result Analysis

The three factor model was studied with the results shown in Table 2. As is shown, the model includes such research indexes as a direct influence, indirect influence and total influence, which is affected by social capital. Then, taking consideration of the mediating influence of pre-decision process and the moderating effect of social capital, the effect of social capital and protective action perceptions of the four factor model was sorted out, with results shown in Table 3 and Table 4.The direct influence, indirect influence, and total influence of social capital have been shown together in Table 4.

#### 3.3.1. The Three Factor Model

Supposing that socioeconomic status had no moderating effect of status, the analysis then evaluated the influence of social capital under the mediating effect of the pre-decision process. In respect of hypothesis H1, all direct influence *T* values of the model were less than 1.96, indicating that the direct influence was not significant. Or in other words, hypothesis H1 is not tenable for the three factor models, with indirect influence generated by social capital. (as shown in Table 2). In terms of hypothesis H2, all indirect influence *T* values of the model were greater than the reference value of 1.96, indicating a significant indirect influence. This proves that the hypothesis H2 is tenable for the three factor model, that is, the pre-decision process does play a mediating effect.

#### 3.3.2. The Four Factor Model

(1) Existence of the Moderating Effect of Socioeconomic Status

In respect of hypothesis H3, all adjustments coefficient *T* values of socioeconomic status were greater than 1.96, which means that the moderating effect was significant. It can be inferred then that the hypothesis H3 for the four factor model is tenable, or in other words, a moderating effect of socioeconomic status does exist.

(2) Influence of social capital with Consideration of Socioeconomic Status

In respect of hypothesis H1, all direct influence *T* values of the three sub-models were less than the reference value of 1.96, indicating that the influence was insignificant, it can be assumed that hypothesis H1 is not tenable for the four factor model. Or in other words, there is an indirect influence of social capital. In respect to hypothesis H2, all indirect influence *T* values of the three sub-models were greater than the reference value of 1.96; the indirect influence can be said to be significant. It can be generated therefore that for the four factor model, hypothesis H2 is tenable; that is, the mediating influence of the pre-decision process does exist.

#### 3.3.3. Comparative Analysis

Table 2 shows that when the moderating effect of socioeconomic status is not considered, the family network, kinship network, and friendship network exert positive impacts on the protective action perceptions; Table 4 shows that when taking the moderating effect of socioeconomic status into consideration, both the kinship network and the family network exert positive influence on the protective action perceptions. The friendship network, on the other hand, exerts a negative influence on protective action perceptions. Finally, the kinds of effects of social capital are reflective of either direct or indirect effects of social capital on the perceptions of protection actions. For example, when the negative indirect effects of social capital were greater than the positive direct effects, the overall impact of social capital on the perception of protection action was negative.

To be more exact, in respect to hypothesis H3, the total influence of the model changed significantly after the moderating variables were included. For example: While the total influence of the family network decreased from 0.486 to 0.245, the total influence of the friendship network also decreased from 0.447 to −0.038. On the other hand, the total influence of the kinship network increased from 0.237 to 0.326.

Because of this, the moderating effect of socioeconomic status on the “friendship network” can be said as directional, that is, when taking socioeconomic status into consideration, the influence of this type of social capital changed from positive to negative; the influence of socioeconomic status on the “kinship network” was enhanced, that is, after considering the socioeconomic status, the influence of this type of social capital became more obvious. The influence of socioeconomic status on “family network” presented as a weakening adjustment, that is, after considering the socioeconomic status, the influence of this type of social capital decreased.

## 4. Discussion

### 4.1. Influence of Social Capital

The following conclusion can be deduced without consideration of the moderating effect of socioeconomic status: the social capital of family network and friendship network had a greater positive influence on the protective action perceptions; the social capital of the kinship network had a less positive influence on protective action perceptions.

As for the social capital of the family network, as long as the individual has enough responsibility for their family, they will seek all types of information which can affect the safety of his/her family members, and take initiatives on reducing the risk of hazardous chemicals faced by his/her family members; Similarly, if the individual acts as the head of the household or plays a influential role in the family, they will have a bigger responsibility and obligation on acquiring more information on the risks of hazardous chemicals. Also, they will have more capabilities of mobilizing more family resources as risk responses for hazardous chemicals; if the family is large-scaled, the responsibility of ensuring the safety of family members will become greater, which will cause the head of the household to become more interested in seeking information on hazardous chemicals and take response measures.

As for the social capital of friendship network: the closer the relationship between individuals and their friends, the higher the frequency for their communication on the information of hazardous chemicals; the more chances individuals have in obtaining useful information; the greater the possibility of one getting help from close friends when dealing with hazardous chemicals. The more deeply individuals trust in their friends, the more confident individuals feel about the information on hazardous chemicals provided by their friends. The greater the heterogeneity of individual friends, the more and more comprehensive information individuals can get from their friends; the greater the possibility for the judgment of individuals affected by contradictory information. The higher the upper reachability, the more and more accurate risk information individuals can get from their friends; the more chances there would be for individuals to get help from friends. The larger the scale of the friendship network, the more chance individuals have in getting more information on hazardous chemicals from their friends.

As for the social capital of kinship network: the regional character of the network not only obstructed individuals from comprehending the extent of risks of hazardous chemicals faced by their long-distance relatives, it has also created fewer chances for the communication on the information of hazardous chemicals between long-distance friends, bringing in less constructive advice. In the meantime, the process of urbanization has kept the geographical location of urban residents and relatives at a distance, causing the relationship between relatives a weak social capital, a less intimate one that is, comparing the relationship between family members or friends. As a result, less information was exchanged, with “trust, emotion and responsibility” of hazardous chemicals kept relatively low. On top of that, the one-child policy has kept the kinship scale relatively small within urban cities, which makes information acquisition on hazard chemicals from relatives less likely. It can be concluded then that the influence of social capital on residents’ protective action perceptions is greater if formed by the family network and is less if the social capital is formed by the kinship network.

### 4.2. Mediating Effect of Pre-Decision Process

From the results, it can be seen that social capital affects protective action perceptions through the pre-decision process. In fact, information itself will not affect people’s behavior. Information needs to go through such a process as being transmitted, received, encoded or decoded before being used by people. In particular, the hazard information on hazardous chemicals is characterized by information asymmetry, technical complexity and stigmatization. This has caused an evidently concealed character for information on potential accidents at the early stages and high redundancy for accident response information at later stages. Even when individuals have obtained relevant information through social capital, the information may not be accurately comprehended. Therefore, the risk information on hazardous chemicals communicated through social capital must be received, and correctly understood by individuals before it can produce any effect on individual protective action perceptions. To let social capital affect more effectively on information acquisition, individuals must be required to have good scientific literacy and information comprehension abilities. This is to ensure that the information brought by social capital can be correctly understood, and will not bring negative influences.

### 4.3. Moderating Effect of Socioeconomic Status

It can be concluded that socioeconomic status makes directional moderation on the path of “friendship network—protective action perceptions”, an enhanced moderation on the path of “kinship network—protective action perceptions”, and a weakening moderation on the path of “family network—protective action perceptions”.

Although the potential risks brought by various hazardous chemical installations or programs make influences on environmental pollution, life and health, and property losses, different individuals would take different response actions. For example, if hazardous chemicals installations are found around the neighborhood, in order to eliminate the hazard source directly, residents with higher socioeconomic status tend to take such specific protective actions as reinforcing, maintaining, moving out, appealing, etc., to reduce or weaken the protective action perceptions. As another example, if the installation of hazardous chemicals is discovered around the community of their relatives, residents with higher socioeconomic status would, out of responsibility, emotion, biological relations, and other reasons, use their own resources or information acquisition abilities to prompt their relatives to take protective actions, which is conducive to an enhancement of protective action perceptions. As another case, residents with higher socioeconomic status, when discovering hazardous chemicals installation near their friends’ residences, because most of their friends also have strong resource and information acquisition abilities, would be overconfident in their friends’ performance in protective actions. Or in other words, the influence of the protective action perceptions is weakening in a negative direction. The role of the moderating effect of socioeconomic status on social capital is to make comprehensive balances on resources, information acquisition and acting ability.

## 5. Conclusions

In order to solve the problem of risk communication on hazardous chemicals, a variable of social capital has been introduced to construct an influence model with data from a questionnaire. Then the following results were obtained: to begin with, while both family networks and friendship networks have greater positive influence, the kinship network has a smaller positive influence; furthermore, social capital exerts influence through the pre-decision process; finally, socioeconomic status makes a directional moderation on friendship networks, an enhanced moderation on kinship networks, and a weakening moderation on family networks. This model should contribute to improving the theoretical system of risk communication based on social relations. The conclusion can be used in making practical plans on risk communication towards residents. Still, more variables should be added to this model in the future.

## Figures and Tables

**Figure 1 ijerph-17-01453-f001:**
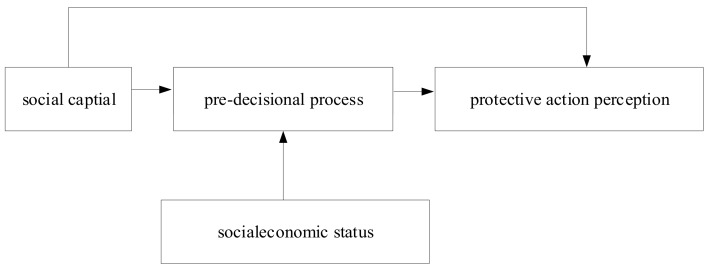
The Influence Model of Social Capital.

**Figure 2 ijerph-17-01453-f002:**
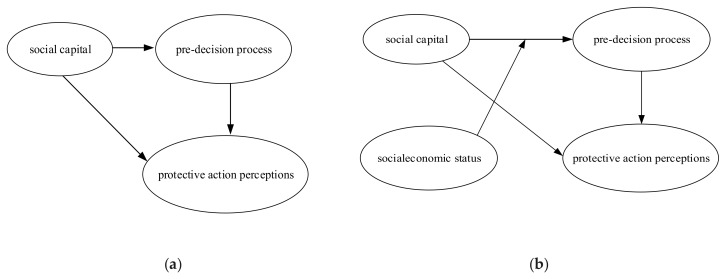
Influence paths of social capital: (**a**) three factor model; (**b**) four factor model.

**Table 1 ijerph-17-01453-t001:** The output values of goodness of fit index.

Social Capital	Model Type	*CMIN*/*DF*	*CFI*	*RMSEA*	*IFI*	*NFI*	*RFI*	*TLI*
Family Network	Three Factor Model	2.030	0.960	0.048	0.961	0.927	0.879	0.935
	Four Factor Model	1.908	0.986	0.045	0.986	0.972	0.986	0.980
Kinship Network	Three Factor Model	0.699	1.000	0.000	1.012	0.974	0.958	1.019
	Four Factor Model	1.124	0.998	0.017	0.998	0.979	0.970	0.997
Friendship Network	Three Factor Model	2.174	0.937	0.051	0.938	0.891	0.842	0.908
	Four Factor Model	1.753	0.982	0.041	0.982	0.959	0.944	0.975

**Table 2 ijerph-17-01453-t002:** Influence without consideration of socioeconomic status.

Social Capital	Model Type	Direct Influence	*T* Value	Indirect Influence	*T* Value	Total Influence
Family network	Three Factor Model	0.265	1.426	0.221	3.652	0.486
Kinship network	Three Factor Model	0.044	0.550	0.193	3.243	0.237
Friendship network	Three Factor Model	0.188	1.257	0.259	3.968	0.447

**Table 3 ijerph-17-01453-t003:** The moderating effect of socioeconomic status.

Social Capital	Model Type	Control Coefficient	*T* Value
Family Network	Four Factor Model	1.055	6.092
Kinship Network	Four Factor Model	1.590	6.028
Friendship Network	Four Factor Model	1.297	7.054

**Table 4 ijerph-17-01453-t004:** Influence with consideration of socioeconomic status.

Social Capital	Model	Direct Influence	*T* Value	Indirect Influence	*T* Value	Total Influence
Family Network	Four factor Model	0.124	0.951	0.121	2.261	0.245
Kinship Network	Four factor Model	0.097	0.756	0.229	3.981	0.326
Friendship Network	Four factor Model	0.116	0.655	−0.154	2.869	−0.038

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
