# Peer review of "The Influence of Social Capital on Protective Action Perceptions Towards Hazardous Chemicals"

_ijerph, 2020, doi:10.3390/ijerph17041453_

Round 1
Reviewer 1 Report
This article studies the interaction of family network, kinship network, and friendship network in processing information about hazardous materials in the Great Bell Temple (GBT) area in Beijing, China, using neighborhood survey method to obtain primary data from the residents.
Conceptually, I think you have to go into greater detail into explaining which one in the household do you choose to survey for, especially when you insist on surveying the household for three times before abandonment, because of inter-household heterogeneity in processing risk-related information between husband and wife, parent and children, etc. This is more nuancedly manifested in the family network variables (3.1.1), in which family responsibility and decision-making power would vastly different between the one responding to the questionnaire (which would probably stay at home during daytime) and the one not available to answering the survey during the field study (which would be earning income for the whole family when out of work).
Overall speaking, I would suggest the authors to stipulate the hypotheses clearly and separately in different paragraphs with indentations, so that the readers can connect the hypothesis from the research question through the results to discussions. In addition, when socioeconomic status is added as a moderating variable, I think the authors should not only vertically compare the moderating effects across different kinds of relationships (i.e., family, kinship, and friendship), but also explain how the direct and indirect influence would reinforce or offset each other in Table 4. Technically speaking, while socioeconomic status reinforces the direct and indirect influence of family and kinship networks (both being positive), the indirect influence more than offsets the direct influence of friendship network so together, socioeconomic status makes friendship network less efficacious in promoting precautious actions on hazardous materials (which, I assume, the authors can go into greater detail into explaining the opposite signs).
An overarching issue on top of my head is whether information requisition and sharing and is without interruptions, because of indispensable legal risk of sharing sensitive information on hazardous materials. Is social capital important in overcoming the legal risk of being muted when sharing “hazardous information” to family, friends and relatives? Or socioeconomic status a hindrance because the more renowned is the survey respondent, the more concerned is that person in promulgating “hazardous information” electronically or else something adverse would happen to his or her personal communication account that explains the negative sign in Table 4? I think for the benefit of the academia and for the sake of advancement of knowledge, these are issues that the authors would take an audacious move in discussing skilfully and objectively.
Besides, I do have some comments and clarifications, most of them lexical and grammatical, as follows:
Line 46: I think it would be more formal to deliberate what the “very few researches” are and how the authors’ paper different from each of these different pieces of prior researches, in order to succinctly highlight the unique contribution of the authors’ present article to the literature.
Line 81: “are on the other ______ related to their own”, which I think an object is missing.
Line 83: Strictly speaking, “hazardous information” is inaccurate because information by itself is not hazardous, but neutral information conveying information about hazardous materials. I think the authors can either define what “hazardous information” is referred to, or paraphrase this as “information conveying hazardous materials”. Same comments for its appearance on line 333.
Line 94: What does the pronoun “their” refer to?
Line 95: lines 35 to 36 say “mainland China”, so does this apply to “China” in line 95 as well?
Line 103: Is geographic distance understood literally as that measured on the map?
Line 112: Please provide a citation for Lindell.
Line 117/149: Should “socioeconomic” be in one word, in line with line 139/304?
Line 147: Either remove “therefore”, or move it to the beginning of this sentence.
Line 148: A hypothesis does not “believe” anything; use “hypothesizes” instead.
Line 170: Please justify the expected percentage to be 80 percent with relevant literature.
Line 176: Please clarify what you mean by “insist on three times”, and how you time the different attempts (e.g., within the same day or every other day/week)? I am curious to learn why it cannot be abandoned, or if you can jump over one household to inquire the next?
Line 182: Do you separate two-year baccalaureate degree from four-year college degree?
Line 185: What kind of requirements do you refer to? Is it a requirement that you set for yourself, or commonly accepted by the literature (if so, please justify with citations).
Line 185: What do you mean by KMO? Please explain any abbreviations that you use.
Line 191: Justify the choice of imputation method you use to deal with missing data.
Line 262: You need a space between “Table 4” and “The Direct” …
Line 271: Grammatically, it should be “proves” instead of “proofs”.
Line 305: I think “changed” is a more precise term for “weakened”.
Line 312: I think the readers are looking for some concrete results, not some conclusion that is probabilistically true as inferred from the phrase “should be”. Please paraphrase this sentence.
Line 317: You don’t need the preposition “on” here which you can remove.
Line 339: Is "long-distance" one word, and is it spatial or psychological? If latter, can you use "distant" or "remote" (if not "unfamiliar") and define it early on?
Line 345: Can you use the one-child policy in lieu of family planning policy when assuming that the former is more commonly known by layman audiences?
Author Response
Open Review#1
Suggestion 1: Conceptually, I think you have to go into greater detail into explaining which one in the household do you choose to survey for, especially when you insist on surveying the household for three times before abandonment, because of inter-household heterogeneity in processing risk-related information between husband and wife, parent and children, etc. This is more nuancedly manifested in the family network variables (3.1.1), in which family responsibility and decision-making power would vastly different between the one responding to the questionnaire (which would probably stay at home during daytime) and the one not available to answering the survey during the field study (which would be earning income for the whole family when out of work).
Response: Thank you very much for the suggestions! The following adjustments were made as a response: first, more content was added concerning the three concepts of family relationship, kinship relationship and friend relationship in 2.1.1 to explains more explicitly the essence of these three types of relationships; second, more details were inserted in 2.3.2 concerning the process of data collections, such as the specific time at which the survey was carried out as well as the intention of researchers to cover all family members (such as husbands and wives, the seniors and the children, etc.). In terms of “we insist three times”, it refers to such situations at which “the survey would be abandoned if no one is available for the survey after contact are made three times.” While the adoption of this practice is restrained by the actual conditions of the survey, the practice has made little impact on the actual results of the analysis. Finally, the sex proportion and age distributions of the respondents are pretty reasonable. (details shown in 2.3.3)
Suggestion 2: Over all speaking, I would suggest the authors to stipulate the hypotheses clearly and separately in different paragraphs with indentations, so that the readers can connect the hypothesis from the research question through the results to discussions. In addition, when socioeconomic status is added as a moderating variable, I think the authors should not only vertically compare the moderating effects across different kinds of relationships (i.e., family, kinship, and friendship), but also explain how the direct and indirect influence would reinforce or offset each other in Table 4. Technically speaking, while socioeconomic status reinforces the direct and indirect influence of family and kinship networks (both being positive), the indirect influence more than offsets the direct influence of friendship network so together, socioeconomic status makes friendship network less efficacious in promoting precautious actions on hazardous materials (which, I assume, the authors can go into greater detail into explaining the opposite signs).
Response: The following adjustments have been made in accordance with your suggestions: first, to make the hypotheses easier to read and understood, all hypotheses were clearly defined in another line. Second, the analysis on the direct and indirect impact on social capital was added in Table 4.
Suggestion 3: An overarching issue on top of my head is whether information requisition and sharing and is without interruptions, because of indispensable legal risk of sharing sensitive information on hazardous materials. Is social capital important in overcoming the legal risk of being muted when sharing “hazardous information” to family, friends and relatives? Or socioeconomic status a hindrance because the more renowned is the survey respondent, the more concerned is that person in promulgating “hazardous information” electronically or else something adverse would happen to his or her personal communication account that explains the negative sign in Table 4? I think for the benefit of the academia and for the sake of advancement of knowledge, these are issues that the authors would take an audacious move in discussing skillfully and objectively.
Response: The above question indicates that the social capital can provide the public with risk information other than those that are obtained through official channel, that is, social capital often advises the public with distorted information. There are sure certain risks about this type of information because these “information” is often groundless, and it is prone to become rumors. However, because of the close-knit relations among families, relatives and friends, the public often ignores the legal risks brought by these kinds of “information”. In the meantime, the development of social media also brings conveniences to the spread of such "information". Besides, as were proven, the social and economic status exert important mediating influence on this trend, that is, the higher the social and economic status people are placed at, the more self-protective awareness and self-protective abilities people acquire, the stronger their abilities in the discrimination of risk information, the more cautious they are when publishing risk information, which can also explain for the emergences of symbols in Table 4.
Suggestion 4: Line 46: I think it would be more formal to deliberate what the “very few researches” are and how the authors’ paper different from each of these different pieces of prior researches, in order to succinctly highlight the unique contribution of the authors’ present article to the literature.
Response: Thank you for your comments. As the original conclusion lacks certain evidences, the content has been adjusted with the additions of more references. Moreover, this study make special contributions on identifying the composition of social capital that affects the risk perceptions of hazardous chemicals and the specific impact path of social capital on risk perceptions.
Suggestion 5: Line 81: “are on the other ______ related to their own”, which I think an object is missing.
Response: Rephrased this sentence.
Suggestion 6: Line 83: Strictly speaking, “hazardous information” is inaccurate because information by itself is not hazardous, but neutral information conveying information about hazardous materials. I think the authors can either define what “hazardous information” is referred to, or paraphrase this as “information conveying hazardous materials”. Same comments for its appearance on line 333.
Response: Replaced the expression with “risk information”.
Suggestion 7: Line 94: What does the pronoun “their” refer to?
Response: the word “their” refers to “social capital”. The sentence was restructured to make clearer of the meaning of the sentence.
Suggestion 8: Line 95: lines 35 to 36 say “mainland China”, so does this apply to “China” in line 95 as well?
Response: The term “mainland China: was used in line 35 to 36 since all anti PX events which were discussed extensively in this study occurred within the territory of mainland China and did not involve Hong Kong, Macao and Taiwan. However, as the three special administrative regions share similar arts and cultural practices with mainland China, the term "China" was used in line 95.
Suggestion 9: Line 103: Is geographic distance understood literally as that measured on the map?
Response: “geographic distance” refers to the physical distance between two difference places.
Suggestion 10: Line 112: Please provide a citation for Lindell.
Response: The Lindell researches were quoted as reference 9 in the paper.
Suggestion 11: Line 117/149: Should “socioeconomic” be in one word, in line with line 139/304?
Response: Replaced all “social economics status” with “socioeconomic status”.
Suggestion 12: Line 147: Either remove “therefore”, or move it to the beginning of this sentence.
Response: The sentence was restructured.
Suggestion 13: Line 148: A hypothesis does not “believe” anything; use “hypothesizes” instead.
Response: The correction was made as advised above.
Suggestion 14: Line 170: Please justify the expected percentage to be 80 percent with relevant literature.
Response: The number “0.8” was changed to “80%” as displayed in relevant literature.
Suggestion 15: Line 176: Please clarify what you mean by “insist on three times”, and how you time the different attempts (e.g., within the same day or every other day/week)? I am curious to learn why it cannot be abandoned, or if you can jump over one household to inquire the next?
Response: “insist on three times” means that “the survey would be abandoned if no one is available for the survey after contact are made three times”. Also, the survey was “carried out from 16:00 to 21:00 on three consecutive weekends”.
Suggestion 16: Line 182: Do you separate two-year baccalaureate degree from four-year college degree?
Response: No, We did not separate “two-year baccalaureate degree” from “four-year college degree” as the two degrees are classified as the same degree level by the Chinese higher education system.
Suggestion 17: Line 185: What kind of requirements do you refer to? Is it a requirement that you set for yourself, or commonly accepted by the literature (if so, please justify with citations).
Response: Included relevant references as advised.
Suggestion 18: Line 185: What do you mean by KMO? Please explain any abbreviations that you use.
Response: The word KMO is used as an indicator to compare both the simple and partial correlation coefficients among variables. The full name for KMO was since added.
Suggestion 19: Line 191: Justify the choice of imputation method you use to deal with missing data.
Response: Relevant references concerning how missing values are handled were added.
Suggestion 20: Line 262: You need a space between “Table 4” and “The Direct” …
Response: Added the space.
Suggestion 21: Line 271: Grammatically, it should be “proves” instead of “proofs”.
Response: Replaced all “proofs” in the paper with “proves”.
Suggestion 22: Line 305: I think “changed” is a more precise term for “weakened”.
Response: The word “changed” was used instead of “weakened”.
Suggestion 23: Line 312: I think the readers are looking for some concrete results, not some conclusion that is probabilistically true as inferred from the phrase “should be”. Please paraphrase this sentence.
Response: The sentence has since been restructured.
Suggestion 24: Line 317: You don’t need the preposition “on” here which you can remove.
Response: Removed this “on”.
Suggestion 25: Line 339: Is "long-distance" one word, and is it spatial or psychological? If latter, can you use "distant" or "remote" (if not "unfamiliar") and define it early on?
Response: The "long-distance" here refers to the spatial distance.
Suggestion 26: Line 345: Can you use the one-child policy in lieu of family planning policy when assuming that the former is more commonly known by layman audiences?
Response: To let more people better understand the article, the “one-child policy” was used instead of “family planning policy”.

Reviewer 2 Report
The manuscript deals with the study of hazardous chemicals. In my opinion this paper merits publication after moderate revision, once the authors take into account and address the points reported below.
The authors should supply the exact removal efficiency. Pollutants degradation (e.g., Applied Catalysis B 248 (2019) 298, Chemical Engineering Journal 379 (2020) 122340, Journal of Cleaner Productions 253(2020)120055). These reports need be commented in the introductions. Recent progress should be compared in Table and closely related referneces should be mentioned in the Intoduction.Author Response
Open Review#2
Suggestion: The manuscript deals with the study of hazardous chemicals. In my opinion this paper merits publication after moderate revision, once the authors take into account and address the points reported below. The authors should supply the exact removal efficiency. Pollutants degradation (e.g., Applied Catalysis B 248 (2019) 298, Chemical Engineering Journal 379 (2020) 122340, Journal of Cleaner Productions 253(2020)120055). These reports need be commented in the introductions. Recent progress should be compared in Table and closely related references should be mentioned in the Introduction.
Response: Increased the number of supporting literatures which make analysis on the threat of hazardous chemicals to the environment and human health.

Reviewer 3 Report
This paper is focused on the influence of social capital on protective action perceptions towards hazardous chemicals and uses a questionnaire survey of the application herein. The subject of the manuscript falls within the scope of the journal and the results of the paper are of sufficiently high impact and global relevance for publication in an international journal. The statistical analysis seems to be highly developed and adequate. The questionnaire and hypotheses are described in general form and the framework of the study is described. Extensive statistical analysis is performed but the scale used to grade the evaluation made by each respondent is not clear and may only be seen in the supplementary file by inspection and inference. Providing a clear description of the scale is important for the reader as it is not evident The interpretations and conclusions are adequate; they are justified by the data and consistent with the objectives. Reviewing the document for minor mistakes and errors should be recommended.
Author Response
Open Review#3
Suggestion: Extensive statistical analysis is performed but the scale used to grade the evaluation made by each respondent is not clear and may only be seen in the supplementary file by inspection and inference. Providing a clear description of the scale is important for the reader as it is not evident. Reviewing the document for minor mistakes and errors should be recommended.
Response: In order to assist readers better understand the data research and the analysis process, explanations concerning "Likert 5-scale" were inserted in 2.3.1 of the questionnaire.

Round 2
Reviewer 1 Report
Hi,
I appreciate the efforts that the authors put in revising this article. I think most of my suggestions have been incorporated, though I do have some comments, mostly lexical, for minor edits.
Line 31: if it was a failure…
Line 56: an even crucial role…
Line 63: do you mean research idea instead of research thought?
Line 64: In this paper…and remove the redundant “s” from capital.
Figure 1: remove the “s” from socioeconomic status
Line 78: should it be perception or perceptions?
Line 82: if there are more than one factors, say “refer”.
Line 84: As aforementioned, you mention perceptions in the title but perception here.
Line 85: you can use comma in lieu of the connective “and” so it does not appear two times. It is also platitudinous to write three nouns consecutively, and you may consider paraphrase this as “an accident involving hazardous chemicals”.
Line 94: I find “what is more” a little bit informal. Consider “specifically” or “in particular”.
Line 108: I see no point in using ‘s here.
Line 116: you need a space between [31,32] and the full-stop.
Line 127: Again, social capital is uncountable.
I think I should stop here and refer the authors to professional editing services, presumably someone whose native language is English, and is experienced in editing journal articles. By and large, I think they have responded to my enquiries, though their responses would be more persuasive if justified and corroborated by related literature(s).
Thank you and I look forward to receiving a revised manuscript.
Author Response
Line 31: if it was a failure…
Response: We have modified the language expression.
Line 56: an even crucial role…
Response: We have modified the language expression as you suggested.
Line 63: do you mean research idea instead of research thought?
Response: Yes, I think the "research idea" you proposed are more accurate.
Line 64: In this paper…and remove the redundant “s” from capital.
Response: We have modified the wrong words.
Figure 1: remove the “s” from socioeconomic status
Response: We have removed the “s” from socioeconomic status.
Line 78: should it be perception or perceptions?
Response: It is better to use "perceptions" here.
Line 82: if there are more than one factors, say “refer”.
Response: We have replaced "refers" with "refer".
Line 84: As aforementioned, you mention perceptions in the title but perception here.
Response: We rechecked and replaced all "protective action perception" in the text with "protective action perceptions".
Line 85: you can use comma in lieu of the connective “and” so it does not appear two times. It is also platitudinous to write three nouns consecutively, and you may consider paraphrase this as “an accident involving hazardous chemicals”.
Response: Thanks for your suggestion, we have replaced "and" with a comma.
Line 94: I find “what is more” a little bit informal. Consider “specifically” or “in particular”.
Response: We have replaced " what is more " with " in particular”.
Line 108: I see no point in using ‘s here.
Response: We have removed the "‘s " here.
Line 116: you need a space between [31,32] and the full-stop.
Response: We added a space between [31,32] and the full-stop.
Line 127: Again, social capital is uncountable.
Response: We have replaced "social capitals" with "social capital”.

Reviewer 2 Report
The revision can be accepted.
Author Response
We further made the following changes to the article.
- We have modified some language expressions in the article. For example, line 56 and line 94.
- We have unified some of the wording in the article, such as "social capital" and "protective action perception".
- The research design is further explained in the sampling section of the article.
- We have modified some language expressions in the text to make the results as clear as possible.
